# Targeting Cannabinoid Receptors: Current Status and Prospects of Natural Products

**DOI:** 10.3390/ijms21145064

**Published:** 2020-07-17

**Authors:** Dongchen An, Steve Peigneur, Louise Antonia Hendrickx, Jan Tytgat

**Affiliations:** Toxicology and Pharmacology, KU Leuven, Campus Gasthuisberg, O&N 2, Herestraat 49, P.O. Box 922, 3000 Leuven, Belgium; dongchen.an@kuleuven.be (D.A.); steve.peigneur@kuleuven.be (S.P.); louise.hendrickx@kuleuven.be (L.A.H.)

**Keywords:** cannabinoid receptor type 1 (CB1) and type 2 (CB2), phytocannabinoids, synthetic cannabinoids, structural analysis, animal venoms

## Abstract

Cannabinoid receptors (CB1 and CB2), as part of the endocannabinoid system, play a critical role in numerous human physiological and pathological conditions. Thus, considerable efforts have been made to develop ligands for CB1 and CB2, resulting in hundreds of phyto- and synthetic cannabinoids which have shown varying affinities relevant for the treatment of various diseases. However, only a few of these ligands are clinically used. Recently, more detailed structural information for cannabinoid receptors was revealed thanks to the powerfulness of cryo-electron microscopy, which now can accelerate structure-based drug discovery. At the same time, novel peptide-type cannabinoids from animal sources have arrived at the scene, with their potential *in vivo* therapeutic effects in relation to cannabinoid receptors. From a natural products perspective, it is expected that more novel cannabinoids will be discovered and forecasted as promising drug leads from diverse natural sources and species, such as animal venoms which constitute a true pharmacopeia of toxins modulating diverse targets, including voltage- and ligand-gated ion channels, G protein-coupled receptors such as CB1 and CB2, with astonishing affinity and selectivity. Therefore, it is believed that discovering novel cannabinoids starting from studying the biodiversity of the species living on planet earth is an uncharted territory.

## 1. Introduction

*Cannabis sativa*, commonly known as marijuana, is a plant which has been used throughout human history to treat a wide variety of ailments, such as pain and anxiety, and for recreational purposes. The first natural product isolated from the cannabis plant and then characterized was the phytocannabinoid cannabinol (CBN) [1], followed by the isolation and pharmacological elucidation of the psychoactive Δ9-tetrahydrocannabinol (Δ9-THC) and the non-euphoric cannabidiol (CBD) [1,2]. The knowledge about the structure and pharmacology of CBN, Δ9-THC and CBD has led to major breakthroughs in our understanding of the effects of this plant. Insights into the mechanism of the action of phytocannabinoids led to the identification of two G protein-coupled receptors (GPCRs), cannabinoid receptors type 1 (CB1) and type 2 (CB2) [3,4]. Consequently, endogenous ligands of cannabinoid receptors, also known as endogenous cannabinoids or endocannabinoids, were identified [2,5]. Amid the recognized endocannabinoids, anandamide (synonym for *N*-arachidonoylethanolamine or AEA) and 2-arachidonoyl glycerol (2-AG) were discovered first [6,7,8]. Thereafter, it became clear that endocannabinoids and cannabinoid receptors are pleiotropic signaling molecules belonging to the endocannabinoid system, which also involves the enzymes that catabolize these compounds [9,10,11,12]. This signaling system has been shown to contribute to re-establishing homeostasis after insults, which highlights the therapeutic opportunities for multiple pathologies, such as pain, inflammation, cardiovascular regulation, metabolic disorders, cancer and neurodegenerative disorders [2,13]. In addition, CB1 and CB2 have been proven to play a crucial role in various bioactivities of phytocannabinoids [14], indicating the significance of cannabinoid receptors for the therapeutic effects of the cannabis plant. These discoveries subsequently inspired the constant generation of a wide variety of synthetic cannabinoids with similar or distinct structures as compared with endo- and phyto-cannabinoids. Simultaneously with the progress made in the medical field to develop selective CB1 or CB2 ligands that can modulate biological functions and treat associated diseases, some synthetic cannabinoids have become problematic in the field of recreational use, such as SPICE and K2 [15].

As a therapeutic target, CB2 has a remarkable advantage over CB1 regarding its expression pattern. CB1 is mainly expressed in the human central nervous system (CNS) (Figure 1), and is the main receptor responsible for the psychotropic effects of Δ9-THC as well as the deleterious psychiatric side effects of drugs targeting CB1 [2,16]. The CB1 inverse agonists rimonabant (SR141716) and taranabant (MK-0364) were developed as anti-obesity drugs, but both produce crippling CNS side effects, such as anxiety, depression, and suicidal ideation [17,18,19]. As a consequence, they were either withdrawn from the market or dropped in clinical trials. In contrast, CB2 is predominantly expressed in peripheral tissues, such as the immune system, where it modulates immunological function, cell migration and cytokine release [16,20] (Figure 1). CB2 expression has also been detected in the brain, albeit to a much lower extent in comparison to the immune system or the level of CB1 expression [16] (Figure 1). Notwithstanding a rather limited expression of CB2 in the peripheral nervous system and the CNS, it is undeniable that the CB2 plays an active role in neurological activities, including nociception and neuroinflammation [21,22]. Some CB2-selective agonists have been developed, showing significant efficacy in *in vitro* assays and in animal models without displaying unwanted psychoactive effects. Examples of such CB2-selective agonists are JWH-015, HU-308 and GW-405833 [23,24,25,26,27]. So far, besides a few phytocannabinoids and their analogs, no other CB targeting drugs have reached the market yet for clinical use. Thus, it is believed that selectively targeting CB2 provides a promising pathway of new drug discovery in the area of natural products for the treatment of a number of disorders while avoiding the severe psychiatric side effects associated with CB1.

Recently, structural determination of CB1/CB2 coupled to the Gi protein has revolutionized our understanding of their structures and functions [28,29], alongside a parallel revolution in the methods for determining structures of cannabinoid receptors [30]. For the past few years, X-ray crystallography has been the method of choice for elucidating CB1 and CB2 structures [30,31,32,33]. Thanks to the advent of higher resolution cryo-electron microscopy (cryo-EM) structures that eliminate the problem of crystal-packing artifacts and generate an ensemble of structures [34], cryo-EM has now become the primary method in order to obtain ligand-bound CB1 or CB2 in the active state coupled to the heterotrimeric G protein complex [28,29]. Therefore, the activation mechanisms of CB1 and CB2 have been revealed [28,29]. This is expected to facilitate the rational structure-based design/discovery of drugs selectively targeting cannabinoid receptors.

In the meantime, some efforts have been made to discover novel cannabinoids over the past few years, leading to the emergence of peptide-type ligands of CB1 and/or CB2 from natural sources, other than the cannabis plant. Examples hereof are hemopressin (Hp) and the related peptides VD-Hpα and RVD-Hpα found in mice, rats or humans, as well as Pep19 derived from peptidyl-prolyl cis-trans isomerase A in humans, showing a variety of *in vivo* pharmacological effects depending on CB1, e.g., antinociception and neuromodulation [35,36]. Moreover, Pep19 did not exhibit CNS side effects in rats [35]. These peptides represent valuable starting points for the development of peptide drugs targeting cannabinoid receptors.

Based on the available literature on cannabinoids, it is evident that natural products have been an important source of CB1 and CB2 ligands. Cannabinoid receptors are one of the primary targets of natural products, since over 600 natural GPCR ligands have been isolated from plants, animals, fungi, and bacteria; they predominantly target aminergic, opioid, cannabinoid, and taste 2 receptors [37]. Among the diverse natural GPCR ligands, nature-derived peptides isolated from bacteria, fungi, plants, and venomous animals are an emerging compound class for GPCR ligand discovery according to published data [37]. Over 50% of nature-derived peptides targeting GPCRs discovered so far originate from animal venoms [37]. Animal venoms contain a true pharmacopeia of peptides acting on molecular targets, e.g., GPCRs, often with impressive affinity and selectivity [38]. Examples of the value of venom peptides in guiding the development of human therapeutics targeting GPCRs include the antidiabetic exenatide (Byetta^®^) from the venomous Gila monster (*Heloderma suspectum*) [38] and the analgesic cobratide (also known as cobratoxin) from the Chinese cobra (*Naja atra*). None of the known venom peptides have been described as ligands of cannabinoid receptors to the best of our knowledge. Therefore, for future perspective, animal venoms can be seen as a promising and yet untapped source to find selective and potent ligands of CB1 and/or CB2. 

In this review, we first briefly introduce the cannabinoid receptors CB1 and CB2, and then discuss the key similarities and diversities of their activation mechanisms based on the structural information obtained by cryo-EM. Furthermore, we provide an overview of the current research status of phytocannabinoids and synthetic cannabinoids that have been shown to be ligands of CB1 and/or CB2. In the light of a natural products perspective, recently emerged novel cannabinoids, i.e., peptide-type ligands of cannabinoid receptors from animal sources, are summarized and the potential of animal venoms as a source of novel cannabinoids is demonstrated. In addition, CB1 or CB2 expression systems that can be used to rapidly screen unlabeled natural products *in vitro* are described in the final section. 

## 2. Cannabinoid Receptors

CB1 is encoded by the gene *CNR1* and consists of 472 amino acids in humans (473 amino acids in rats and mice) [16]. The amino acid sequence identity among these species is 97–99% [16]. CB2 is encoded by the gene *CNR2*, which consists of 360 amino acid in humans. It shares only 44% sequence homology with CB1 at the protein level [16]. Additionally, CB2 has greater species differences between humans and rodents, compared with CB1, as its amino acid sequence identity among humans and rodents is slightly higher than 80% [16]. 

CB1 and CB2 are both class A (rhodopsin-like) GPCRs. Generally, the structure of cannabinoid receptors contains seven transmembrane alpha helices (TMHs) arranged to form a closed bundle and loops connecting TMHs that extend intra- and extracellularly [34]. In addition, it contains an extracellular N terminus and an intracellular C terminus that begins with a short helical segment (Helix 8)-oriented parallel to the membrane surface [34]. CB1 primarily couples to Gi/o protein and, under certain conditions, couples to Gs and Gq, while CB2 only couples to Gi/o [34], to trigger the further activation and downstream signaling.

## 3. Ligand-Bound CB1/CB2-Gi Complex

### 3.1. Activation Mechanism of CB1 and CB2

Ligand-bound cryo-EM structures of the active cannabinoid receptors in complex with Gi (Figure 2A,C) have recently been built and utilized to reveal activation mechanisms of CB1 and CB2. The overall structures of the active CB1-Gi and CB2-Gi complexes are alike [29]. The binding poses of agonists in CB1 and CB2 are superimposable [28,29]. Moreover, the agonist-binding pockets and conformations of critical residues for the receptor activation are almost identical between the active conformations of CB1 and CB2 [29].

On the other hand, intriguing differences in the activation process of both cannabinoid receptors have also been revealed by structural analysis. Firstly, in the case of agonist-bound cannabinoid receptor-Gi complexes, the cytoplasmic region of transmembrane region 5 (TM5) in CB1 is simply extended and moves inward during activation, resulting in more polar and hydrophobic interactions with α5 of the Gαi protein [29]. In contrast, the cytoplasmic portion of the TM5 in CB2 extends and moves outward to form extensive interactions with the α5 helix of Gαi [29]. Secondly, in CB1, TM6 in the intracellular region moves inward to interact with α5 of the Gαi protein [29]. However, a large outward movement of the intracellular part of TM6 in CB2 occurs to accommodate the mounting of α5 from the Gαi protein [29]. In addition, the residues on the cytoplasmic ends of TM5 and TM6 in CB2 shift modestly upward, relative to those of CB1 [28]. The movements of TM5 and TM6 have a certain significance in the activation processes of cannabinoid receptors: an outward movement of TM6 is suggested as a characteristic of cannabinoid receptor activation and an extension of TM5 can result in new interactions with Gαi [29]. Notably, the critical so-called ‘‘toggle switch’’ residues of cannabinoid receptors show differences for both receptors. In CB1, the “twin toggle switch”, F200 and W356 (Figure 2B), experiences synergistic conformation changes, while in CB2, the “single toggle switch’’, W258 (corresponding to W356 in CB1) (Figure 2B), triggers the activation and the downstream signaling [29]. Alternatively, when taking into consideration residue F117 in CB2 (corresponding to F200 in CB1), another way to think about the differences of the toggle switch in the cannabinoid receptors is the following: the distance between F200 and W356 in CB1 is longer than that between F117 and W258 in CB2 (Figure 2D) [28]. This is a result of the upward position of W356 and the rotation of F200 in CB1 compared with the analogous residues in CB2 [28]. The different arrangement of the toggle switch in CB2 causes a rotation of F202 in its TM5 in comparison with the corresponding L287 in CB1 [28]. In general, CB2 only experiences minor conformational changes upon agonist binding, while CB1 is exceptional and displays larger conformational changes when modulated by agonists [29]. Furthermore, the high plasticity of CB1 during the transitions between different states facilitates its inherent ability to respond to a diverse array of ligands compared to CB2 [29].

### 3.2. Implications for CB1 and CB2 Ligand Selectivity

Structure determination of cannabinoid receptors coupled to Gi indicates that discovering a selective agonist may be a huge challenge due to similar binding pockets in both cannabinoid receptors. Nevertheless, there is clear-cut evidence that some highly selective CB1 or CB2 agonists can be obtained. For example, arachidonyl-2’-chloroethylamide (ACEA) and arachidonylcyclopropylamide (ACPA) are potent and selective CB1 agonists, as described further in “4.2.2. CB1-selective agonists”; JWH-133, JWH-015 and PM-226 are potent and selective CB2 agonists, as described further in “4.2.3. CB2-selective agonists”. This strongly suggests that the assumed critical differences of activation processes between CB1 and CB2 might be a good starting point for the design of cannabinoid receptor-selective drugs. As we emphasized earlier, the difference of agonist-binding activation modes between CB1-Gi and CB2-Gi complexes may find its basis in the “toggle switch”. Although the agonist-bound CB1-Gi and CB2-Gi overlap very well, the “toggle switch” concept in cannabinoid receptors is regarded as a crucial role in determining efficacy of a ligand [28,29]. When the notorious switch is constrained by binding of an antagonist or inverse agonist, the activation of cannabinoid receptors can be blocked or reversed [28]. 

## 4. Phytocannabinoids and Synthetic Cannabinoids

### 4.1. Phytocannabinoids

To date, over 100 unique phytocannabinoids have been identified [14,39]. These phytocannabinoids can be classified into several subclasses, including the Δ9-THC type, the Δ8-THC type, the CBD type, the CBN type, and several others [14]. Among these, Δ9-THC and CBD are the compounds that have been investigated the most [40]. They have been shown to bind to cannabinoid receptors and elicit the characteristic beneficial or psychoactive effects associated with cannabis. The beneficial effects of the phytocannabinoids are mediated by multiple targets, thus not solely the cannabinoid receptors, CB1 and CB2, as many people believe [14]. Two kinds of novel phytocannabinoids have recently emerged (Δ9-THCP and CBDP) [39] (see further). Here, we provide an overview on the recently reported novel phytocannabinoids and the well-known/typical phytocannabinoids that are the most thoroughly studied to date. 

#### 4.1.1. Δ9-Tetrahydrocannabinol (Δ9-THC)

The psychotropic effects of cannabis are considered to be produced essentially by Δ9-THC (Table 1a); for example, acute psychotic reactions and a temporary decline in cognitive functioning in human [1,2,40,41]. The psychoactive effects of cannabis are predominantly attributed to partial agonist activity of Δ9-THC at CB1 [14,40] (Table 1a). Moreover, Δ9-THC is also characterized as a partial agonist of CB2 [14,42] (Table 1a). As a typical partial agonist, Δ9-THC has a mixed agonist–antagonist effect which is presumably dependent on the cell type, the expression of receptors, and the presence of endocannabinoids or other full agonists [14]. Regarding the unwanted side effects, the safety concerns raised in connection with Δ9-THC as a psychoactive agent preclude its widespread use in the clinic. Δ9-THC undoubtedly has a range of important therapeutic benefits, such as appetite stimulation, analgesia, and anti-emetic effects, mediated by either CB1 and CB2 or non-cannabinoid targets [40,43]. This drove further large-scale investigations, leading to the approval of nabiximol (Sativex^®^), a combination of THC and CBD, for the treatment of pain and/or spasticity in multiple sclerosis which was a milestone in cannabis research [44]. Sativex^®^ is a preparation administered in the form of an orally mucosal spray and licensed in more than 27 countries as a formulation delivering a consistent concentration at a one-to-one ratio of Δ9-THC:CBD [45]. After the optimization of preparation and delivery methods, another product, called Cannador^®^, came to the market, delivering Δ9-THC:CBD within a narrow concentration range and at a two-to-one ratio, in the form of an orally administered capsule [45]. In relation to this, many of the recent studies on medical cannabis have focused on various forms of Δ9-THC + CBD administration [46,47,48,49] and co-administration of Δ9-THC with first-line neurotherapeutic drugs [50]. On the other hand, the synthesis of Δ9-THC analogs is another effective way to avoid or reduce its severe side effects. Details of synthetic cannabinoids are summarized further in the “Synthetic cannabinoids” section. 

#### 4.1.2. Cannabidiol (CBD)

Unlike Δ9-THC, CBD (Table 1b) is regarded as a clinically interesting compound for its therapeutic potential in several disorders, including anti-inflammatory, analgesic, anti-anxiety, and antitumor properties [40,43,51,52]. Moreover, it has low addictive, hallucinogenic, and toxic side effects [40,43,51]. A number of studies have investigated CBD to determine its activity at cannabinoid receptors and shown very low affinity for CB1 and CB2 [14] (Table 1b). It was reported that CBD can act as an antagonist/inverse agonist at certain concentrations below which it binds to both CB1 and CB2 orthosteric sites [53]. Recently, several studies demonstrated that CBD can act as a negative allosteric modulator of CB1, which alters the potency and efficacy of the orthosteric ligands but does not activate the receptor itself [54,55,56]. For CB2, the study showed that CBD can act as a partial agonist [54]. These results could explain the reported ability of CBD to functionally antagonize some undesirable effects of Δ9-THC in animal studies and clinical studies in humans without attenuating positive effects, thus increasing the therapeutic index of Δ9-THC. In addition, CBD exerts analgesic effects in rats by interacting with several target proteins including CB1 which involves in nociceptive control [57]. In addition, the potential immunological or anti-inflammatory effects of CBD are likely mediated via CB2 [43]. In other words, the mechanistic bases of the effects of the phytocannabinoid CBD are still not fully explained. CBD is also reported to be a potent ligand of transient receptor potential vanilloid 1 (TRPV1) and TRP melastatin 8 (TRPM8) channels [58]. As CBD shows significant efficacy as a therapeutic agent with broad safety, the U.S. Food and Drug Administration (FDA), in June 2018, approved the first drug Epidiolex^®^. This is an oral solution composed of the active ingredient CBD, derived from marijuana, to treat rare and severe forms of epilepsy. The proof of concept delivered by Epidiolex^®^ drives further research formulating CBD in order to apply CBD to other various diseases or to improve the efficacy of other medical drugs in co-administration [59,60].

#### 4.1.3. Δ9-Tetrahydrocannabiphorol (Δ9-THCP) and Cannabidiphorol (CBDP)

At the end of last year, two novel phytocannabinoids, Δ9-tetrahydrocannabiphorol (Δ9-THCP) and cannabidiphorol (CBDP), were isolated from *Cannabis sativa* [39]. These common names were derived from the traditional naming of phytocannabinoids based on the resorcinyl residue, in this case corresponding to sphaerophorol [39]. Δ9-THCP is a Δ9-THC homolog with a seven-term side alkyl chain (Table 1c) which is longer than the five-term side alkyl chain of Δ9-THC (Table 1a). Δ9-THCP can bind with high affinity to both CB1 and CB2 in a radioligand binding assay [39] (Table 1c). Its affinity for CB1 is significantly higher compared to the reported data of Δ9-THC, as shown in Table 1a. Further *in vivo* evaluation of Δ9-THCP confirmed its cannabimimetic activity of decreasing locomotor activity and rectal temperature, inducing catalepsy and producing analgesia, thereby mimicking the properties of a full CB1 receptor agonist [39]. The cannabimimetic activity of Δ9-THCP is several times higher than that of Δ9-THC [39]. As the pharmacological activity of Δ9-THC is particularly ascribed to its affinity for CB1 receptor, it is suggested that this affinity can be increased by elongating the alkyl side chain [63]. Thus, the *in vivo* results of Δ9-THCP show the significance of the length of the side alkyl chain on the resorcinyl moiety in modulating the ligand affinity at CB1 [39]. Another novel phytocannabinoid was named cannabidiphorol (CBDP), which is a CBD homolog with a seven-term side alkyl chain (Table 1d). At present, no data on the pharmacological effects of CBDP are available [39].

### 4.2. Synthetic Cannabinoids

Synthetic cannabinoids constitute the most diverse group of cannabinoids in regard to functional profile and chemical structure [40]. Originally, the synthetic cannabinoids were used as pharmacological tools to delineate the cannabinoid receptor-mediated activity [21]. Thus, their structural characteristics allow them to bind to one of the known cannabinoid receptors present in human cells, CB1 and/or CB2 [15]. After decades, some synthetic cannabinoids emerged on the market as alternatives to phytocannabinoids for recreational drug use. Several hundreds of different synthetic cannabinoids have been produced up to date, sometimes with subtle structural changes [15,22]. These synthetic cannabinoids can be divided into classical, nonclassical, amino-alkylindoles, eicosanoids and others in terms of chemical structure [53]. Many of these synthetic cannabinoids are used in pharmacological studies involving structure–activity relationships, receptor binding studies and detailed mechanisms of action of these drugs. The FDA has approved three synthetic cannabis-related drug products: Marinol^®^ (dronabinol), Syndros^®^ (dronabinol), and Cesamet^®^ (nabilone) [64]. Marinol^®^ and Syndros^®^ include the active ingredient dronabinol, a synthetic Δ9-THC which is considered the psychoactive intoxicating component of cannabis (i.e., the component responsible for the “high” people may experience from using cannabis). Their therapeutic uses in the United States include the treatment of nausea associated with cancer chemotherapy and the treatment of anorexia associated with weight loss in AIDS patients [64]. Another FDA-approved drug, Cesamet^®^, contains the active ingredient nabilone, which has a chemical structure similar to THC and is synthetically derived. Cesamet^®^, similarly to dronabinol-containing products, is indicated for nausea associated with cancer chemotherapy and neuropathic pain [64]. 

#### 4.2.1. Mixed CB1/CB2 Agonists

Most known synthetic agonists of cannabinoid receptors show little selectivity between CB1 and CB2 [21], but exhibit stronger affinity for cannabinoid receptors compared to endo- and phytocannabinoids [65]. The synthetic cannabinoids that are most commonly used in the laboratory as CB1 and CB2 receptor agonists fall essentially into three chemical groups: classical, nonclassical and amino-alkylindole. Three notable examples of such compounds are 11-hydroxy-∆8-THC-dimethylheptyl (HU-210), CP-55,940 and WIN-55,212-2. HU-210 (Table 2a), an example of a classical synthetic cannabinoid, is a highly potent cannabinoid receptor agonist, and its potency and affinity at cannabinoid receptors exceed those of many other cannabinoids [66]. The high potency and affinity of HU-210 are believed to result from replacing the pentyl side chain on Δ9-THC with a dimethylheptyl group [53]. Furthermore, pharmacological effects of HU-210 *in vivo* are exceptionally long-lasting [66]. The non-classical synthetic cannabinoid, CP-55,940 (Table 2b), is a cannabinoid receptor full agonist that is considerably more potent than Δ9-THC [67,68]. Moreover, it has comparable affinity for both CB1 and CB2 receptors in the low nanomolar range and it is highly potent *in vivo* [69]. Like CP-55,940, the amino-alkylindole synthetic cannabinoid WIN-55,212 (Table 2c) exhibits relatively high potency for both CB1 and CB2, and possesses CB1 and CB2 affinities in the low nanomolar range. However, in contrast to CP-55,940, it has slightly greater affinity for CB2 than for CB1 [66] (Table 2c). Overall, agonists of the cannabinoid receptors are involved in cognition, memory, anxiety, control of appetite, emesis, motor behavior, sensory, autonomic and neuroendocrine responses, immune responses and inflammatory effects, liver injury and hepatocellular carcinoma [53]. Currently, among synthetic CB1/CB2 mixed agonists, only nabilone (Table 1d) is in the phase III of the clinical trial for non-motor symptoms in Parkinson’s disease from ClinicalTrials.gov [70].

#### 4.2.2. CB1-Selective Agonists

The starting point for the development of a CB1-selective agonist was the AEA molecule. Through changing the atom on the 1′, 2′ or 2 carbon of AEA, its CB1 selectivity can be enhanced, leading to synthesis of CB1-selective agonists, such as methanandamide (Table 3c) and O-1812 (Table 3d) [66]. So far, the most potent CB1-selective agonists developed are arachidonyl-2´-chloroethylamide (ACEA) (Table 3a) and arachidonylcyclopropylamide (ACPA) (Table 3b), both of which exhibit reasonably high CB1 potency. ACEA displays nanomolar affinity at CB1 and >1000-fold selectivity over CB2 [21], while ACPA displays >300-fold selectivity over CB2 [74]. In general, compounds with an agonistic effect and sufficient affinity to CB1 have a potential for abuse as cannabis substitutes. However, considering the medical potency of CB1-selective agonists, there is still an attractive interest in those compounds exploring different pharmacological strategies to minimize the unwanted CNS side effects and maximize the beneficial therapeutic effects. For example, ACEA and ACPA have both been studied and have been shown to have anti-depressant [82,83] and anti-nociceptive effects [84,85]. Furthermore, ACEA has been receiving considerable attention in terms of co-administration with different antiepileptic drugs. It potentiated the anticonvulsant activity of antiepileptic drugs in various animal models of epilepsy and stimulated neurogenesis in the brain of mice, showing no possible acute adverse effects [86,87,88,89,90]. This combinational activity could be beneficial to avoid the severe side effects of CB1-selective agonists. However, no CB1-selective agonist is currently in the stage of the clinical trial according to ClinicalTrials.gov [70].

#### 4.2.3. CB2-Selective Agonists

Concerning CB2-selective agonists, the most widely used experimental tool is the classical cannabinoid, JWH-133 (Table 4a), and the less selective amino-alkylindole, JWH-015 (Table 4b), developed by Dr John Huffman [66]. Both compounds not only bind with higher affinity to CB2 than to CB1, but also behave as a potent CB2-selective agonist in functional assays [66]. Other notable CB2-selective agonists include PM-226 (Table 4c), HU-308 (Table 4d), the GlaxoSmithKline compound GW-405833 (Table 4e), Merck Frosst (now known as Merck Canada) compounds L-759,633 (Table 4f) and L-759,656 (Table 4g). CB2-selective agonists have undoubtedly been the focus in the field of therapeutic uses, because modulation of the CB2 is an interesting approach avoiding CNS related side effects, to treat pain, inflammation, arthritis, addictions, neuroprotection, and cancer, among other possible therapeutic applications [24,27,95,96,97,98,99,100]. Interestingly, the use of known CB2-selective agonists (i.e., JWH-015 and L-759,656) for treating or preventing a disease associated with immune dysfunction such as HIV disease was proposed in an US patent published in 2012 [22]. Over the past decade, published patents have claimed >150 synthetic selective agonists of CB2 [22]. Nowadays, new ligands designed to interact with CB2 as selective agonists are currently the subject of research both by academia and industry. Furthermore, a number of reports dealing with *in vivo* and *in vitro* models have shown positive and very interesting results (as summarized in Table 4). Nevertheless, there has still been limited success in clinical trials, partly due to the lack of translation from preclinical models and also due to the differences across species [22,101,102]. At present, at least three unique synthetic CB2 agonists have reached clinic trials, including GW842166X, S-777469 and JBT-101 from ClinicalTrials.gov [22,70,103].

#### 4.2.4. CB1-Selective Antagonists/Inverse Agonists

Since the discovery of CB1 and the subsequent development of the CB1-selective and potent antagonist SR141716 (also called Rimonabant) (Table 5a) by Sanofi-Aventis [19,116], there has also been considerable interest in the therapeutic potential of CB1-selective antagonists. Researchers found their therapeutic potential in the treatment of disorders in which the endocannabinoid system appears to induce undesirable symptoms following its upregulation [1]. Other notable CB1 selective antagonists include analogs of rimonabant, AM-251 (Table 5b) and AM-281 (Table 5c) [117]. Rimonabant, AM-251 and AM-281 not only act as antagonists attenuating effects of CB1 agonists, but act as inverse agonists which can by themselves elicit responses in some CB1-containing tissues that are opposite in direction from those elicited by CB1 agonists [117]. More specifically, they appear to produce inverse cannabimimetic effects in at least some tissues by somehow reducing the constitutive activity of CB1. The constitutive activity is understood as the coupling of CB1 to its effector mechanisms that, it is thought, can occur in the absence of exogenously added or endogenously released CB1 agonists [117]. Since the withdrawal of rimonabant from the market in 2008, due to its severe psychiatric side effects, research on CB1-selective antagonists’ potential pharmacological effects has continued. For instance, a recent study showed that rimonabant protects against light-induced retinal degeneration *in vitro* and *in vivo* via regulating CB1 [118]. More recently, it was shown to exhibit neuroprotective effects in a retinal degeneration model by blocking CB1 [119]. At the same time, as a selective antagonist/inverse agonist of CB1, implications of rimonabant in weight loss, anti-diabetes and reduced drug dependency have been established [53]. However, although research on the development of synthetic CB1-selective antagonists sounded very promising, it remains associated with unideal convoys. Several compounds have been withdrawn from commercial markets and clinical trials [120]. Besides rimonabant, taranabant (MK-0364) (Table 5d) and otenabant (CP-945,598) (Table 5e) were both discontinued in phase III clinical trials for treating obesity due to the risk/reward ratio [17,121] and surinabant (SR147778) (Table 5f) was discontinued from clinical trials for smoking cessation [53]. Moreover, there is no CB1-selective antagonist which is now in the stage of the clinical trial according to ClinicalTrials.gov [70]. Therefore, the current strategy towards tackling these adverse effects may be to restrict binding of CB1 antagonists to CB1 in CNS, limit their crossing of the blood brain barrier, or co-administer CB1 antagonists with drugs blocking side effects.

#### 4.2.5. CB2-Selective Antagonists/Inverse Agonists

The most notable CB2-selective antagonists/inverse agonists are the Sanofi-Aventis diarylpyrazole, SR144528 [135] (Table 6a) and 6-iodopravadoline (AM-630) [115] (Table 6b). Both compounds bind with much higher affinity to CB2 than to CB1, exhibit marked potency as CB2-selective antagonists and behave as inverse agonists that can produce inverse cannabimimetic effects at CB2 by themselves [66]. In fact, less attention has been paid to CB2-selective antagonists/inverse agonists compared to agonists, as indicated by a small number of patents and pharmacological studies over the past few years. For example, the US patent for AM-630 describes this compound as a CB2-selective antagonist and proposes the use of AM-630 for treating or preventing a disease associated with immune dysfunction such as HIV disease [22]. In addition, AM-630 has been shown to effectively inhibit inflammatory osteolysis in the differentiation medium system [136] and to potentiate the activity of conventional antidepressant drugs *in vivo* [26]. In addition, in contrast to CB2-selective agonists, no CB2-selective antagonist has been in the stage of clinical trial so far from ClinicalTrials.gov [70].

#### 4.2.6. Allosteric Modulators

Allosteric ligands are ligands binding to a site topographically different from the orthosteric binding site. They can be divided into three groups according to their effects on the orthosteric ligand (bind to the same site as the endogenous ligand) responses: positive allosteric modulators (PAMs), negative allosteric modulators (NAMs), and neutral allosteric ligands (NALs). They hold great therapeutic potential, as they do not possess intrinsic efficacy. Instead, they enhance or diminish the response of orthosteric ligands at receptors allowing for the tempering of cannabinoid receptor signaling without the desensitization, tolerance, and dependence [103]. In addition, allosteric modulators of cannabinoid receptors have numerous advantages over the orthosteric ligands, such as a higher receptor type selectivity, probe dependence and biased signaling. Therefore, they have a great potential to separate the therapeutic benefits from side effects specifically seen with some orthosteric ligands [103]. The notion of an allosteric modulator of CB1 was first reported by Price et al. in 2005 [138]. They found three allosteric modulators of CB1, Org29647 (Table 7a), Org27759 (Table 7b) and Org27569 (Table 7c) which were shown to enhance the binding of the CB1 agonist CP-55,940, while reducing the binding of the antagonist/inverse agonist SR141716. Among them, Org27569 has the best PAM effect on CP-55,940 binding [103]. Although Org27569 acts as PAM for binding, it reduces CP-55,940-dependent G protein-coupling, producing an insurmountable antagonism of the receptor functionality and inhibits CP-55,940-dependent cAMP inhibition, consistent with a behavior as NAM [103,138]. Since then, other synthetic CB1 allosteric modulators were developed, such as the PAMs GAT211 (Table 7d), ZCZ011 (Table 7f), RTI-371 (Table 7g), and the NAM PSNCBAM-1 (Table 7h). Like Org27569, PSCBAM-1 shows a PAM profile in binding assays but behaves as a non-competitive functional antagonist, decreasing the cellular response induced by orthosteric agonists [139]. Among these CB1 allosteric modulators, the synthetic indole Org27569 is one of the most intensively studied. Recently, the crystal structure of the ternary complex of CB1 with orthosteric agonist CP-55,940 and the allosteric molecule Org27569 were reported for the first time, illustrating a potential strategy for the drug modulation of CB1 and other class A GPCRs [140]. To date, a number of Org27569 or PSNCBAM-1 analogs and other types of CB1 allosteric modulators have been developed in order to improve specificity [12]. However, only two CB1 allosteric modulators, ZCZ011 [141] and the racemic mix GAT211 [142,143,144], have been demonstrated to elicit *in vivo* activity through a CB1 mechanism of action. Interestingly, PAM activity of GAT211 *in vivo* was recently shown to reside with its S-(-)-enantiomer (GAT229) (Table 7e), constituting the first demonstration of enantiomer-selective CB1 positive allosteric modulation in animal models [143]. Contrarily, none of the other CB1 allosteric modulators’ CB1-depending *in vivo* effects have been demonstrated to date [12,103]. For example, Org27569 has been shown not to alter *in vivo* pharmacological effects (i.e., antinociception, catalepsy, and hypothermia) of orthosteric CB1 agonists (i.e., AEA, CP-55,940, and Δ9-THC) and not to alter the discriminative stimulus effects of AEA in fatty acid amide hydrolase-deficient mice [12]. In conclusion, it remains important to demonstrate the *in vivo* effects of candidate compounds producing their allosteric actions at CB1. 

Compared to CB1, only two synthetic allosteric modulators have been developed for CB2 so far. One of them is cannabidiol-dimethylheptyl (CBD-DMH or HU-219) (Table 7i) which has been shown to act as an allosteric modulator for CB2, reducing CP-55,940 binding and being a PAM of cAMP modulation but a NAM of β-arrestin1 recruitment [54]. In addition, CBD-DMH can be considered as a mixed agonist/PAM of CB1 [54]. The other CB2 allosteric modulator is the compound C2 (Table 7j), which can act as a CB2 PAM in binding assays and displays antinociceptive activity *in vivo* in an experimental mouse model of neuropathic pain [145]. Apparently, all CB allosteric modulators are still in the stage of preclinical studies instead of the clinical trial according to ClinicalTrials.gov [70].

## 5. Novel Cannabinoids from Animal Sources

GPCRs mediate a series of signal processes in the human body [150]. One third of clinically used drugs target these receptors [150,151]. Considering that many of the natural endogenous ligands of GPCRs are peptides (comprising 50 or fewer amino acids), drug development based on these peptides is suggested as an attracting and promising way to produce drugs for the treatment of diseases in which the GPCR signaling pathway plays a pivotal role [150]. Inspired by the successful medicinal use of the endogenous peptide insulin, researchers began to focus on research and the development of peptide-type drugs [151]. A milestone of the research on natural endogenous peptides is the discovery of hemopressin (Hp) and its derivates RVD-Hpα and VD-Hpα which have been proven to be ligands of cannabinoid receptors involving numerous physiological and pathological processes [152,153]. These peptides can be regarded as novel cannabinoids because they exhibit activity at cannabinoid receptors and have distinct structures from phytocannabinoids and synthetic cannabinoids. Their actions at cannabinoid receptors and the associated pharmacological effects have attracted considerable attention since their discovery, leading to many studies showing satisfying results *in vitro* and *in vivo*. A few years ago, a novel peptide Pep19 derived from peptidyl-prolyl cis-trans isomerase A in humans was reported to act as a CB1 ligand and showed surprising *in vivo* results that are related to CB1 [35]. In this part, we briefly introduce the reported peptides targeting CB1/CB2. Additionally, from a perspective point of view, the potential of venom peptides as a source for novel cannabinoids is demonstrated, since they currently account for the largest portion of discovered nature-derived peptides targeting GPCRs [37]. 

### 5.1. Peptides Targeting CB1/CB2

#### 5.1.1. Hemopressin and its Derivates VD-Hpα and RVD-Hpα

Hemopressin (Hp or rat Hp, sequence: PVNFKFLSH) (Table 8a) and related peptides VD-Hpα (pepcan-11, sequence: VDPVNFKLLSH) (Table 8b) and RVD-Hpα (pepcan-12, sequence: RVDPVNFKLLSH) (Table 8c) are endogenous peptides that have been found to bind to cannabinoid receptors. Among them, Hp was first discovered and identified in rat brain and spleen [153]. Thereafter, VD-Hpα and RVD-Hpα were identified in mouse brain as well as mouse and human plasma [152]. It was found that they are fragments of α-hemoglobin in these animals. Because CB1 is primarily expressed in the CNS, these peptides were investigated as endogenous ligands modulating CB1 activity. 

Hp was originally identified as an inverse agonist/antagonist of CB1 [154]. A few years ago, a study revealed that Hp slightly activated G proteins and functioned as a very weak agonist, proposing a hypothesis that Hp indirectly interacts with CB1 [155]. In addition, it is confirmed that Hp does not function as a NAM of CB1 in a study using electrophysiology [156]. 

In contrast to Hp, VD-Hpα and RVD-Hpα were initially suggested to be agonistic ligands of CB1 [152]. Later on, RVD-Hpα was reported as a potential NAM of CB1 in the brain [157]. Soon after, it was identified as a PAM of CB2 in peripheral tissues [158]. Moreover, RVD-Hpα was demonstrated to exhibit the effect of CB2-mediated G-protein recruitment and cAMP inhibition without affecting β-arrestin-2 recruitment and receptor internalization [158]. 

Some pharmacological effects of Hp, VD-Hpα and RVD-Hpα related to cannabinoid receptors have also been characterized. Hp and the extended peptides VD-Hpα and RVD-Hpα can cause CB1-mediated neuromodulation. This effect was shown in a study which demonstrated that Hp blocked CB1 agonist-mediated neurite outgrowth in Neuro-2A cells (a mouse neuroblastoma cell line with neuronal and amoeboid stem cell morphology) expressing CB1 in a manner similar to that of SR141716 [154], while VD-Hpα and RVD-Hpα promoted neurite outgrowth in a manner similar to that of the CB1 agonist HU-210 in Neuro-2A cells. This effect could be inhibited by pretreatment with the antagonist SR141716 [152]. In addition, Hp acts as an endogenous functional antagonist of CB1 and modulates the activity of appetite pathways in the brain. It was shown that Hp dose-dependently decreased night-time food intake in normal male rats and mice, as well as in obese male mice, without causing any obvious side effects [159]. The anorectic effect was absent in CB1 null mutant male mice, and Hp could block CB1 agonist CP-55,940-induced hyperphagia in male rats, providing strong evidence for antagonism of CB1 *in vivo* [159]. Moreover, Hp was shown to reduce hepatic collagen deposition, downregulate CB1 and CB2 expression, and increase matrix metalloproteinase-1 expression by the *per os* administration of Hp for 2 weeks [160]. Consequently, Hp alleviated liver fibrosis in the bile duct-ligated rats in comparison with the CB2 agonist β-caryophyllene. This could be partly attributed to its antagonism/inverse agonism at CB1 [160]. Regarding VD-Hpα, it was proven to possess antinociceptive effects mediated by CB1. It is a dose-dependent antinociceptive effect achieved at the levels of the spinal cord [161]. This antinociceptive effect can be blocked by the CB1 antagonist AM-251 but not the CB2 antagonist AM-630 or by naloxone, suggesting that the mechanism is related to CB1 rather than CB2 or opioid receptors [161]. Furthermore, VD-Hpα can inhibit gastrointestinal transit via the activation of CB1 located in the brain, because VD-Hpα via intracerebroventricular administration inhibits upper gastrointestinal transit and colonic expulsion [162]. These effects are prevented by AM-251 instead of AM-630 [162]. 

Notably, the side effects of Hp, VD-Hpα and RVD-Hpα cannot be ignored. Intracerebroventricular administration of VD-Hpα or RVD-Hpα impairs the memory of normal mice [163]. In the case of Hp, the side effects so far found are of psychiatric nature. Hp administration induced anxiogenic and depressive behavior, decreased monoamine steady state levels in prefrontal cortex, and increased the gene expression of the enzymes involved in the catabolism in rats [164]. The mechanism of these side effects has not been completely clarified.

#### 5.1.2. Pep19

In 2017, Reckziegel et al. reported a novel peptide Pep19 (Sequence: DIIADDEPLT) (Table 8d) exhibiting inverse agonistic activity at CB1 [35]. Pep19 is a peptide resulting from the rational modification of the original peptide (sequence: DITADDEPLT) which is derived from peptidyl-prolyl cis-trans isomerase A in humans. The inverse agonistic activity of Pep19 is greatly enhanced compared with the original peptide which shows very weak inverse agonistic activity at CB1 [35]. Oral administration of Pep19 to rats improved several metabolic parameters, including a reduction in the serum glucose, triacylglycerol and blood pressure, without changing heart rate, as well as reducing the whole adiposity index and the mass of gonadal and mesenteric adipose tissues [35]. Moreover, oral administration of Pep19 significantly increased the expression of uncoupling protein 1 in specific cells of the inguinal adipose tissue [35]. This effect was proven again in both white adipose tissue and 3T3-L1 differentiated adipocytes, and was blocked by AM-251, a CB1 antagonist [35]. Surprisingly, Pep19 did not exhibit cellular toxicity or typical undesired CNS effects related to cannabinoid receptors in rodents, even after 10 days of chronic treatment with Pep19 [35]. Although Pep19 was shown to work similarly to Hp in this study, reducing body weight and improving metabolic parameters, it has the advantage of lacking the undesired side effects associated with CB1 inverse agonists such as rimonabant and Hp [35].

### 5.2. Potential of Animal Venoms as a Source for Novel Cannabinoids

Amid the vast animal sources on planet Earth, animal venoms are regarded as a tremendous treasure-house of venom toxins (i.e., venom peptides) that specifically, potently, stably and speedily manipulate physiological targets including ion channels and receptors. Animal venoms, like parts of the plant *Cannabis sativa*, have been used to treat some ailments throughout human history. Since the 7th century BCE, snake venom has been used in Ayurvedic medicine to prolong life and treat arthritis and gastrointestinal ailments, while tarantulas are used in the traditional medicine of indigenous populations of Mexico and Central and South America [38]. In the 1970s, an active peptide was found in the venom of the Brazilian snake *Bothrops jaracaca*. On this basis, the blockbuster drug captopril was developed. It is an inhibitor of the angiotensin converting enzyme and is now used clinically as an antihypertensive drug. Since then, drug discovery based on animal toxins has increased significantly. Moreover, the advancement of technologies has enabled the study of venoms from animals that are small, rare, or hard to maintain in the lab, which greatly facilitates the high-throughput screening of animal venoms and the characterization of venom peptides’ structure and function [38]. The research on venom-derived drugs has so far yielded at least 10 registered and deposited drugs on the market [165], examples of which include captopril (Capoten^®^), tirofiban (Aggrastat^®^) and eptifibatide (Integrilin^®^). Although humans have been using venoms for thousands of years, only for the past five or six decades scientists have been studying the venoms at the molecular level using modern biochemistry, physiology and biophysics and other technologies [38]. So, enormous strides in the research of animal toxins have been made over the past half a century [38].

Venom peptides, e.g., from cone-snails, snakes, spiders and scorpions, are currently of particular interest as a source of lead compounds for the development of GPCR ligands among naturally occurring compounds, since they cover a chemical space, which differs from that of synthetic small molecules [37]. They are recognized as reliable alternatives for small molecules, owing to their higher selectivity [166]. In addition, peptides may be metabolized and cleared without accumulation in body tissues, thereby minimizing the occurrence of side effects [167]. Many GPCR-targeting venom-derived peptides were discovered to be beneficial in diverse pathological animal models [37]. For example, mambaqauretin-1 was recently isolated from the venom of a green mamba (*Dendroaspis angusticeps*), exhibiting high affinity and inhibitory action on the vasopressin type 2 receptor [168]. Further *in vivo* experiments highlighted the potential usefulness of mambaquaretin-1 for the treatment of polycystic kidney disease [168]. Exenatide derived from exendin-4 which was isolated from the venom of the Gila monster (*Heloderma suspectum*) is an example of a GPCR-targeting nature-derived peptide drug (Byetta^®^) as described in the “Introduction”. It was introduced in 2005 to treat diabetes mellitus type 2 and acts as an agonist of the glucagon-like peptide-1 receptor [38]. Another example is the National Medical Products Administration of China (NMPA)-approved drug cobratide, derived from venoms of Chinese cobra (*Naja atra*). This is an analgesic peptide targeting the acetylcholine receptor. Interestingly, δ-CNTX-Pn1a, a peptide from *Phoneutria nigriventer* spider venom (also called PnTx4(6-1) or δ-Ctenitoxin-Pn1a), has been reported to induce antinociception in rat models of pain, involving both opioid and cannabinoid systems [169]. It was also reported that peripheral interactions between opioid and cannabinoid systems contribute to antinociception of a peptide, crotalphine, of which the sequence is based on the structure of the natural analgesic factor isolated from the venom of the South American rattle-snake *Crotalus durissus terrificus* [170]. Moreover, PnPP-19 derived from venom of the spider *Phoneutria nigriventer* was shown to induce central and peripheral antinociception through both opioid and cannabinoid systems in *in vivo* pain models [171,172]. This peptide displayed long-lasting analgesic activity after oral administration in rats [173]. However, the detailed action mechanisms of these peptides on opioid or cannabinoid receptors remain unknown. In the search for novel compounds that produce analgesia via GPCR modulation, animal venoms offer an enormous and mostly untapped source of potent and selective peptide molecules [174]. Another example related to this is that the GABA_B_ receptor (GABA_B_R) appears to play a critical role in analgesia of α-conotoxin Vc1.1 in rodent models of pain, while the precise nature of α-conotoxin Vc1.1 binding to GABA_B_R is currently unknown [174]. Additionally, it is evident that cannabinoid receptors are one of the primary GPCR targets of natural products based [37]. Thus, targeting the GPCRs, i.e., cannabinoid receptors, shows a great prospective potential for medical uses of animal venoms which indeed represent a promising and uncharted pharmacopeia.

## 6. Expression Systems of Cannabinoid Receptors Used for Screening Natural Products *In Vitro*

The classical expression system of cannabinoid receptors used for rapidly analyzing the binding and function of phytocannabinoids, synthetic cannabinoids and novel peptide-type cannabinoids *in vitro* uses mammalian cells transfected with human CB1 or CB2, e.g., Chinese hamster ovary (CHO) cells and embryonic mouse fibroblast (3T3-L1 cells) [14,35,157,158]. These mammalian transient expression systems enable the flexible and rapid production of membrane-bound cannabinoid receptor proteins. They are ideal for the expression of these human proteins because these systems generate recombinant proteins with more native folding and post-translational modifications—such as glycosylation—than expression systems based on hosts such as *E. coli*, yeast, or insect cells. In addition, certain cultured cell lines that express cannabinoid receptors naturally, and cannabinoid receptor-containing membrane preparations obtained from tissues (such as brain and spleen), are also commonly used [61].

The *Xenopus laevis* oocyte transfected with human CB1 or CB2 and the accessory G protein-activated inwardly rectifying potassium (GIRK) channels 1/2 or GIRK1/4 as well as regulator of G-protein signaling 4 (RGS 4) is also a robust and reliable system for screening unlabeled natural products. The direct electrophysiological measurement of transmembrane K^+^ currents is an effective method to study the effects of compounds on membrane-bound cannabinoid receptors. As shown in Figure 3, a typical trace of K^+^ current evoked by the cannabinoid receptor agonist WIN-55,212-2 (WIN) (0.5 μM) is shown. An advantage of the *Xenopus* oocytes is that they are able to properly assemble and incorporate functionally active proteins, e.g., cannabinoid receptors, into their plasma membranes, which is the reason that the *Xenopus* oocytes are widely used as a heterologous expression system [175]. Using this system, it is also possible to functionally investigate the membrane protein CB1 or CB2 in combination with its accessory proteins, in order to screen potential drugs [175]. Another advantage of using the heterologous *Xenopus* oocyte expression system is the freedom to co-express different combinations of membrane-bound proteins: one can, for instance, co-inject mRNA encoding CB1 and TRPV1 (the vanilloid receptor, also the target of some cannabinoid ligands), or CB2 and mu opioid receptors, in order to check the possible promiscuity of ligands, or CB1 with different types of voltage-gated ion channels (Na, K, Ca, Cl). Furthermore, it also allows researchers to carry out structure–function research, based on the expression of mutant CB1/CB2. As such, this functional bioassay can unravel hitherto unknown coupling(s) of signal-transmission pathways, e.g., the cannabinoid and opioid pathways, allowing us to better understand the analgesic properties of some cannabinoids. It is notable that this system also nicely mimics the physiological reality of CB1 or CB2. This is because these receptors have been shown to couple to GIRK channels [176]. The activation of the cannabinoid receptors affects GIRK channels [177], so that they can exert the inhibitory regulation of neuronal excitability in most brain regions and heart rate [176]. Neuronal GIRK channels are predominantly heteromultimers composed of GIRK1 and GIRK2 subunits in most brain regions, whereas atrial GIRK channels are heteromultimers composed of GIRK1 and GIRK4 subunits [176].

In addition, when using these expression systems, it is also important to pay attention to a phenomenon called shear stress, acting on the cell and which can rupture the fragile membrane, thereby releasing the intracellular material. Since there is no cell wall in animal and mammalian cells, they rely on a plasma membrane to keep the intracellular contents stay intact.

## 7. Conclusions and Perspectives

Starting from the research on the medicinal cannabis plant, hundreds of different phyto- and synthetic cannabinoids targeting CB1 and/or CB2 have emerged with diverse pharmacological effects, for either medicinal or recreational use. CB1 and CB2, as the earliest-identified targets of the phytocannabinoids in the human body, have received considerable attention in the field of therapeutics. In general, agonists of the cannabinoid receptors are theoretically important for ameliorating neurological or brain diseases, treating pain and inflammation as well as many cancers. On the other hand, antagonists/inverse agonists have been shown to primarily play a role in weight loss, diabetes and treating feeding disorders. However, to date, both natural and synthetic ligands of the cannabinoid receptors show only a few clinical applications. This indeed requires more mechanistic investigations in a systematic way in order to delineate the mode of action of the phytocannabinoids and synthetic cannabinoids. In this case, the ligand-bound cryo-EM structures of the active cannabinoid receptors in complex with G-protein are necessary and could assist in identifying more drug leads, especially in light of the withdrawals often occurring for many drugs targeting cannabinoid receptors. The activation mechanisms of CB1 and CB2 have been recently revealed thanks to the powerfulness of cryo-EM, which may enable rational, structure-guided cannabinoids design. According to the European Monitoring Centre for Drugs and Drug Addiction (EMCDDA), over 620 new psychoactive substances (NPS) are currently being monitored through the EU Early Warning System. One hundred and sixty-nine of these are synthetic cannabinoid receptor agonists, with 14 recognizable chemical families. More and more ligands of CB1 or CB2 with new cannabinoid scaffolds showing potential therapeutic activity *in vivo* have emerged over the past decade [78,178,179,180,181]. The structures of many synthetic cannabinoids can be categorized into four components: tail, core, linker and linked group. Assigning each component, a code name allows the chemical structure of the cannabinoid to be identified without the long chemical name. From all the possible combinations, it is clear that structure−activity relationships (SAR) of these compounds is almost endless, hard to oversee and that many more NPS compounds will continue to emerge [182].

The novel cannabinoids—peptides—from animal sources draws a brighter future for therapeutically targeting of CB1/CB2. The four peptide-type ligands of cannabinoid receptors found so far have shown potential therapeutic effects depending on CB1 *in vivo*, and one of them has shown no CNS side effects. For future perspectives, it is promising to discover more novel cannabinoids with less or without CNS side effects from animals living on planet Earth. In particular, animal venoms contain a true pharmacopeia of venom peptides targeting ion channels or receptors with incredible selectivity, potency and speed. This makes them promising and ideal drug leads. Venomous animal species account for ~15% of all described animal biodiversity on Earth, yet we know very little about their venoms [38]. For rapidly determining the effects of the natural products for CB1 and/or CB2 *in vitro*, the robust cannabinoid receptor expression systems mimicking their physiological reality are commonly relied on. Hence, we are convinced that there will be more exciting discoveries in the future. The field of the development of venom drugs targeting CB1/CB2 is starting a new chapter.

## Figures and Tables

**Figure 1 ijms-21-05064-f001:**
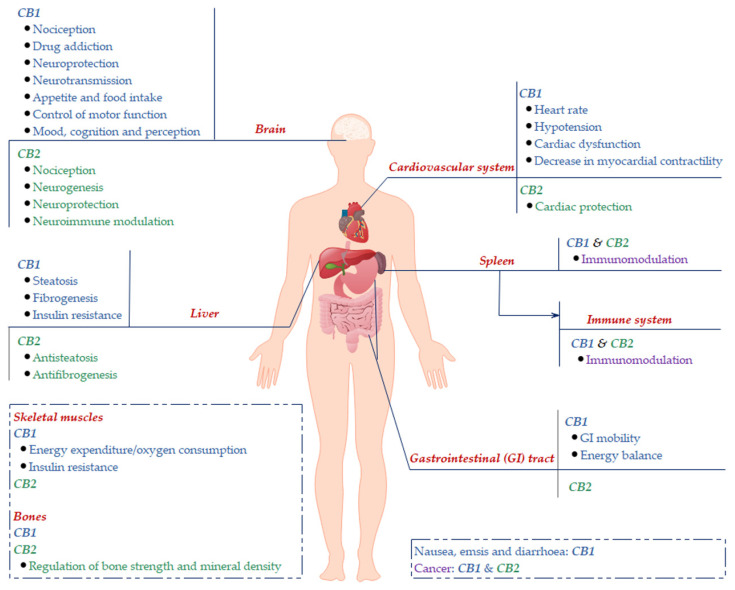
Major localization of CB1 and CB2 and their associated physiology in the human body. CB1: the majority of CB1 was found to be expressed in the brain, where it modulates various neurological activities. CB1 is located in the peripheral tissues, and, although to a lesser extent, also participates in the modulations of local tissue functions. CB2: the predominant expression of CB2 was revealed to be in the immune system (such as the spleen), where it exhibits the effects of immune modulation, and other peripheral tissues.

**Figure 2 ijms-21-05064-f002:**
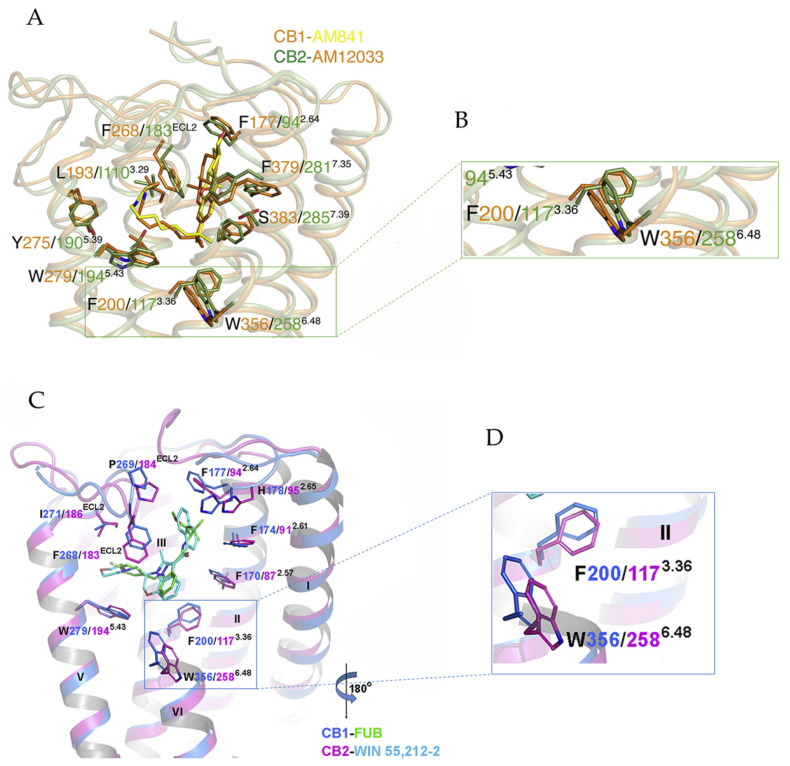
(**A**) Binding pocket of human CB1-Gi complex bound to agonist AM841 and human CB2-Gi complex bound to agonist AM12033. Orange cartoon, CB1 structure; green cartoon, CB2 structure; yellow sticks, AM841; orange sticks, AM12033. (**B**) Comparison of the “toggle switch” residue conformation in human CB1, orange cartoon; and in human CB2, green cartoon. (**C**) Binding pocket of human CB1-Gi complex bound to agonist FUB and human CB2-Gi complex bound to agonist WIN 55,212-2. Marine cartoon, CB1 structure; purple cartoon, CB2 structure; green sticks, FUB; cyan sticks, WIN 55,212-2. (**D**) Comparison of the “toggle switch” residue conformation in human CB1, marine cartoon; and in human CB2, purple cartoon. (**A**) and (**B**) are derived from Hua et al. [29], (**C**) and (**D**) are derived from Xing et al. [28].

**Figure 3 ijms-21-05064-f003:**
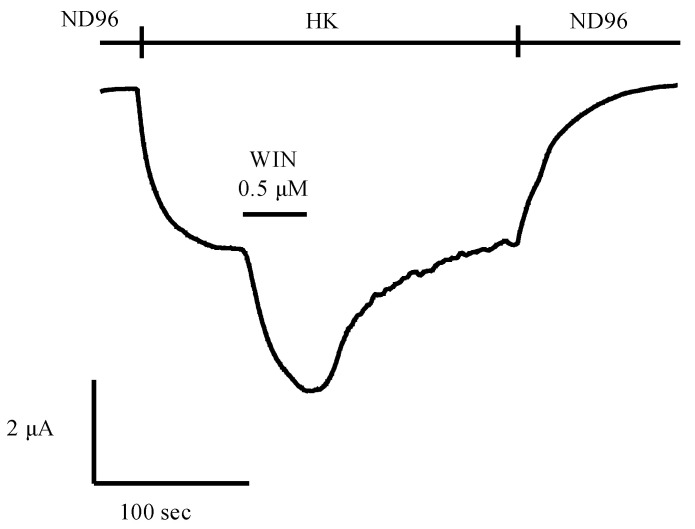
Cannabinoid agonist WIN-55,212-2 (WIN) activated inwardly rectifying potassium currents in oocytes co-expressing GIRK1/2 + RGS4 with CB1. Currents were induced by exchanging ND96 with HK while the oocytes were voltage-clamped at −90 mV. Current enhancement observed on application of 0.5 μM WIN in the presence of HK. In the final, oocytes were perfused with ND96. The experiment was repeated in at least 4 oocytes. In the case of CB2, a similar approach is followed. ND96: physiological salt buffer solution (96 mM NaCl, 2 mM MgCl_2_, 2 mM KCl, 5 mM HEPES, and 1.8 mM CaCl_2_, with a final pH of 7.5), HK: high potassium solution for the measurement of K^+^ currents through GIRK1/2 channels (96 mM KCl, 2 mM NaCl, 1 mM MgCl_2_, 1.8 mM CaCl_2_, 5 mM HEPES with a final pH of 7.5).

**Table 1 ijms-21-05064-t001:** Structures, binding type and bioactivities of Δ9-THC, CBD, Δ9-THCP and CBDP.

Phytocannabinoids	Binding Type/CB	K_i_ (nM)/CB	EC_50_/IC_50_ (nM)/CB	Bioactivity
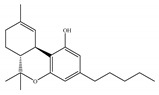 (a) Δ9-THC	Partial agonist/CB1, CB2	5.00~80.0/CB1 1.70~75.0/CB2 [14]	13.0~87.0/CB1 41.8, 61.0/CB2 [14]	Analgesic, antiemetic, orexigenic [53]; relief from muscle spasms/spasticity in multiple sclerosis [61]
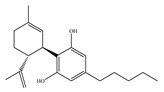 (b) CBD	Antagonist/inverse agonist, negative allosteric modulator/CB1 Partial agonist/CB2	73.0~>10,000/CB1 370~>10,000/CB2 [14]	3860/CB1 503, 2270/CB2 [14]	Anti-inflammatory, anti-nociceptive, anti-oxidant, anti-ischemic, neuroprotective, immunosuppressive [62]; anxiolytic [43,62]
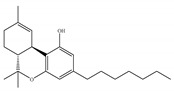 (c) Δ9-THCP	Agonist/CB1, CB2	1.20/CB1 6.20/CB2 [39]	NA	Analgesic [39]
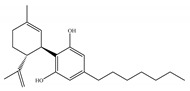 (d) CBDP	NA	NA	NA	NA

K_i_: binding constant; EC_50_: half-maximal effective concentration; IC_50_: half-maximal inhibitory concentration; NA: not available.

**Table 2 ijms-21-05064-t002:** Structures, chemical type and bioactivities of mixed CB1/CB2 agonists.

Synthetic Cannabinoids	Chemical Type	K_i_ (nM)/CB	EC_50_ (nM)/CB	Bioactivity
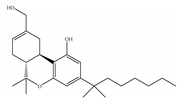 (a) HU-210	Classical	0.0608~0.730/CB1 0.170~0.524/CB2 [66]	0.0702/CB1 [71] NA/CB2	Analgesic [53]; neuroprotective [72,73]
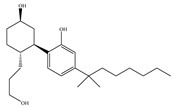 (b) CP-55,940	Nonclassical	0.500~5.00/CB1 0.690~2.80/CB2 [66]	0.0462~31.0/CB1 [24,71,74] NA/CB2	Anti-nociceptive, anti-emetic [53]
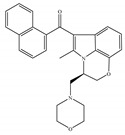 (c) WIN-55,212-2	Amino-alkylindole	1.89~123/CB1 0.280~16.2/CB2 [66]	5.50~3000/CB1 [71,74,75,76,77,78] NA/CB2	Analgesic, anti-inflammatory [53]; neuroprotective [79]
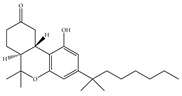 (d) Nabilone	Classical	1.84/CB1 2.19/CB2 [66]	NA	Analgesic, antiemetic, anti-inflammatory [53]; neuroprotective [80,81]

K_i_: binding constant; EC_50_: half-maximal effective concentration; NA: not available.

**Table 3 ijms-21-05064-t003:** Structures, chemical type and bioactivities of CB1-selective agonists.

Synthetic Cannabinoids	Chemical Type	K_i_ (nM)/CB	EC_50_ (nM)/CB1	Bioactivity
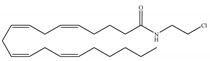 (a) ACEA	Eicosanoid	1.40, 5.29/CB1 195, >2000/CB2 [66]	0.0317, 51.0 [74]	Anti-depressant [86]; anti-nociceptive [53]; anti-ulcer [84]; neuroprotective [91,92]; potentiating activity of antiepileptic drugs [87]; reducing cognitive impairment [91]
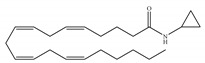 (b) ACPA	Eicosanoid	2.20/CB1 715/CB2 [66]	0.0551, 37.0 [74]	Anti-depressant, anxiolytic [83]; anti-nociceptive [53];
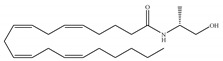 (c) Methanandamide	Eicosanoid	17.9~28.3/CB1 815~868/CB2 [66]	1000 [77]	Analgesic, anti-emetic, orexigenic, anti-proliferation, anti-migration [53]
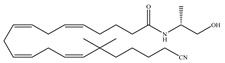 (d) O-1812	Eicosanoid	3.40/CB1 3870/CB2 [66]	NA	Anti-nociceptive, suppressing spontaneous activity and catalepsy [93]; anti-convulsant [94]

K_i_: binding constant; EC_50_: half-maximal effective concentration; NA: not available.

**Table 4 ijms-21-05064-t004:** Structures, chemical type and bioactivities of CB2-selective agonists.

Synthetic Cannabinoids	Chemical Type	K_i_ (nM)/CB	EC_50_ (nM)/CB2	Bioactivity
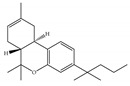 (a) JWH-133	Classical	677/CB1 3.40/CB2 [104]	63.0 [105]	Attenuating neurodegenerative and spatial memory impairment [27]; improving cerebral infarction [106]; anti-inflammatory, ameliorating sepsis [107]; anti-cancer [53]; anti-nociceptive [108]; protective effects on renal ischemia-reperfusion injury [109] and against cardiotoxicity [110]
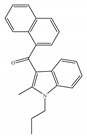 (b) JWH-015	Amino-alkylindole	383/CB1 13.8/CB2 [104]	NA	Attenuating neurodegenerative, neuroprotective [106]; immunomodulatory, anti-inflammatory [53]; anti-nociceptive [111]; anti-cancer [112]; anti-obesity [25]
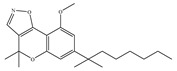 (c) PM-226	Classical	>40,000/CB1 12.8/CB2 [95]	38.7 [95]	Neuroprotective [95]; anti-neuroinflammatory [78]
** 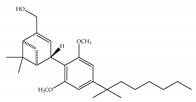 **(d) HU-308	Nonclassical	>10,000/CB1 22.7/CB2 [66]	5.57 [23]	Anti-convulsant, neuroprotective [106]; anti-inflammatory [96,97]; anti-nociceptive [96]; anti-dyskinesia [98]; osteoprotective [97]; alleviating septic lung injury [113]
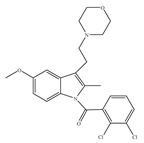 (e) GW-405833	Amino-alkylindole	4772/CB1 3.92/CB2 [24]	0.650 [24]	Anti-nociceptive, anti-inflammatory [99]; protective effects on acute liver injury [114]
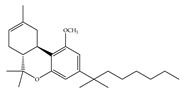 (f) L-759,633	Classical	1043, 15850/CB1 6.40, 20.0/CB2 [66]	8.10 [115]	Analgesic [53]
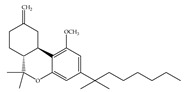 (g) L-759,656	Classical	529~>20000/CB1 11.8~57.0 [66]	3.10 [115]	Analgesic [53]

K_i_: binding constant; EC_50_: half-maximal effective concentration; NA: not available.

**Table 5 ijms-21-05064-t005:** Structures, chemical type and bioactivities of CB1-selective antagonists/inverse agonists.

Synthetic Cannabinoids	Chemical Type	K_i_ (nM)/CB	IC_50_ (nM)/CB1	Bioactivity
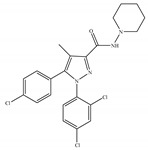 (a) Rimonabant (SR141716)	Others	1.80~12.3/CB1 702~13200/CB2 [66]	5.60~48.0 [116,122]	Anti-obesity, smoking cessation [53]; protective effects of retinal degeneration [118]; neuroprotective [119]
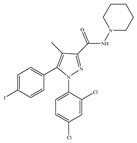 (b) AM-251	Others	7.49/CB1 2290/CB2 [66]	3.00 [122]	Anti-obesity; anti-depressant [53]; potentiating activity of antidepressant drugs [123]; improving albuminuria and renal tubular structure [124] as well as recognition memory [125]
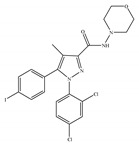 (c) AM-281	Others	12.0/CB1 4200/CB2 [66]	9.91 [122]	Improving cognitive deficits [53]; facilitatory effect on recognition memory [126]; ameliorating spatial learning and memory impairment [127]; protective effects against cardiotoxicity [110]
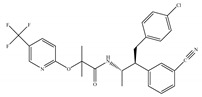 (d) Taranabant (MK-0364)	Others	0.130, 0.270/CB1 170, 310/CB2 [104]	0.290 [128]	Anti-obesity [129]; smoking cessation [130]; anti-nociceptive [131];
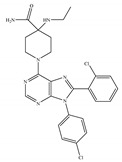 (e) Otenabant (CP-945,598)	Others	0.120~0.700/CB1 [132,133] 7663/CB2 [132]	13.1 [122]	Anti-obesity [120]
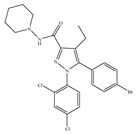 (f) Surinabant (SR147778)	Others	3.50/CB1 442/CB2 [134]	9.60 [134]	Anti-obesity, smoking cessation, suppressing alcohol preference [53]

K_i_: binding constant; IC_50_: half-maximal inhibitory concentration; NA: not available.

**Table 6 ijms-21-05064-t006:** Structures, chemical type and bioactivities of CB2-selective antagonists/inverse agonists.

Synthetic Cannabinoids	Chemical Type	K_i_ (nM)/CB	IC_50_ (nM)/CB2	Bioactivity
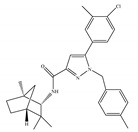 (a) SR144528	Others	50.3~>10,000/CB1 0.280~5.60/CB2 [66]	39.0 [135]	Anti-nociceptive [53]
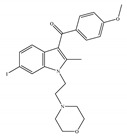 (b) AM-630	Amino-alkylindole	5152/CB1 31.2/CB2 [66]	12.3 [122]	Inhibiting inflammatory osteolysis [136]; potentiating activity of antidepressant drugs [26]; improving memory, anti-oxidant [137]

K_i_: binding constant; IC_50_: half-maximal inhibitory concentration; NA: not available.

**Table 7 ijms-21-05064-t007:** Structures, chemical type and bioactivities of allosteric modulators.

Synthetic Cannabinoids	Chemical Type	Bioactivity
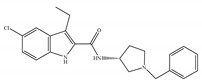 (a) Org29647	Amino-alkylindole	NA
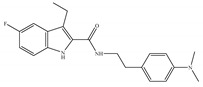 (b) Org27759	Amino-alkylindole	NA
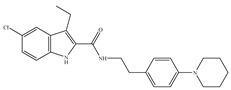 (c) Org27569	Amino-alkylindole	Anti-obesity [12]
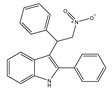 (d) GAT211	Amino-alkylindole	Reducing the signs and symptoms of Huntington’s disease [146]; augmenting the pharmacological effects of the CB1 orthosteric agonists [12]; anti-nociceptive, anti-inflammatory [142]
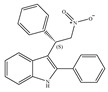 (e) GAT229	Amino-alkylindole	Anti-nociceptive and anti-inflammatory in combination with the orthosteric CB1 agonist [143]; reducing the signs and symptoms of Huntington’s disease [146]
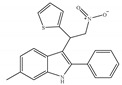 (f) ZCZ011	Amino-alkylindole	Anti-nociceptive, anti-inflammatory [139]
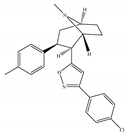 (g) RTI-371	Others	Blocking cocaine-induced locomotor stimulation [147]
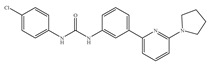 (h) PSNCBAM-1	Others	Anti-obesity [139]
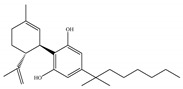 (i) CBD-DMH (HU-219)	Nonclassical	Anti-inflammatory [148], anxiolytic [149]
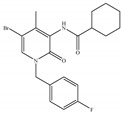 (j) Compound C2	Others	Anti-nociceptive [145]

NA: not available.

**Table 8 ijms-21-05064-t008:** Sequences, bioactivities and sources of novel peptide-type cannabinoids.

Peptides	Sequence	Bioactivity	Source
(a) Hemopressin (Hp)	PVNFKFLSH	Neuromodulatory, anorexigenic, alleviating liver fibrosis, antinociceptive [36]	α-Hemoglobin in rat
(b) VD-Hpα (pepcan-11)	VDPVNFKLLSH	Neuromodulatory, antinociceptive, inhibiting gastrointestinal mobility, inducing tolerance to thermal antinociception, orexigenic [36]	α-Hemoglobin in human and mouse
(c) RVD-Hpα (pepcan-12)	RVDPVNFKLLSH	Neuromodulatory, anxiolytic, anti-depressant [36]	α-Hemoglobin in human and mouse
(d) Pep19	DIIADDEPLT	Reducing body weight, improving metabolic parameters [35]	Peptidyl-prolyl cis-trans isomerase A in human

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
