# Peer review of "Targeting Cannabinoid Receptors: Current Status and Prospects of Natural Products"

_ijms, 2020, doi:10.3390/ijms21145064_

Round 1
Reviewer 1 Report
Cannabinoid receptors (CB1 and CB2) play a critical role in numerous human physiological and pathological conditions. In this review, the author provided a comprehensive summary on the structure and modulators of Cannabinoid receptors. In the reviewer’s perspective, the review of this field is necessary and useful for chemists and biologists to efficiently discover better CB modulators for disease treatment. Thus, this manuscript is suitable for publishing in Int. J. Mol. Sci. after the following concerns to be addressed.
- The activity data of compounds should be included in the tables, such as the Ki and EC50 value.
- The description for synthetic CB modulators (section 4.2) was two rough and missed a lot of papers, such as JMC, 2010, 53, 16, 5915-5928; 2009, 52, 19, 5785-5788; 2013, 56, 14, 5722-5733; 2016, 59, 14, 6753-6771; 2018, 61, 4, 1646-1663. The reviewer suggests the authors to re-organize this section and provide comprehensive summarize on all of the representative compounds. In addition, it will be better if the author could draw a table to provide more detailed information about the compounds, including the chemical structure, chemical type, biological data, development stage and bioactivity. Moreover, the authors should also provide a brief SAR to elucidate the discovery process of every representative compound.
- The drawing for some chemical structures are unprofessional, such as table 4, L-759.
Reviewer 2 Report
This appears to be an excellent review, being highly accessible and understandable to readers within and outside the field. It appears to provide a well-balanced and comprehensive assessment of natural products that can modulate these receptors.
Minor comments.
1. The manuscript including the title includes a large number of possessive nouns, which are often discouraged by most publishers. Consider rephrasing each of these or at least most of these. Further the term "natural product's perspective" is infrequently used in publication and is an anthropomorphism and should be revised including in the title.
2. I am unsure how much post-editing MDPI does, but the font of the legend of Figure 1 varies as does Table 8 and should be standardized.
3. Line 127: change "sequence homology" to "sequence identity".
4. Figure 2 legend should include which species of cannadinoid receptors are represented. Also, potential copyright issues with Figure 2 need to be considered and possibly copyright/permissions obtained.
5. Line 303: Insert space after "[69]".
6. Line 465: Given Ref. 36 is form 2017, change "A couple of years" to "A few years" or "Three years".
7. Line 484: Change truncation "doesn't" to "does not" and any others as identified.
8. Lines 643-645: This paragraph has no context and in its current form is unclear and there appears to be some lead sentence missing. Please revise accordingly.
9. Figure 3: It is unclear what ND96 is; please clarify in legend. Should HK and ND96 be added to Abbreviations list (page 23)?
10. Referencing not consistent. For example many DOis are absent (but appreciate not all articles have a DOI); Ref. 1 "PNAS" but elsewhere (Ref. 146) in full; not all journal correctly capitalized (an annoying feature of more recent versions of Endnote I suspect); Please revise and standardize.
Round 2
Reviewer 1 Report
All the concerns has been adressed and well-responsed, and this manuscript is suitable for beening published Int. J. Mol. Sci. However, the author should double check the drawing of compounds in the tables before publishing. It seems the structure of L-759,633 and 656 are incorrect.
